# Molecular Mechanism of Tanshinone against Prostate Cancer

**DOI:** 10.3390/molecules27175594

**Published:** 2022-08-30

**Authors:** Wei Li, Tao Huang, Shenghan Xu, Bangwei Che, Ying Yu, Wenjun Zhang, Kaifa Tang

**Affiliations:** Department of Urology, The Affiliated Hospital of Guizhou Medical University, Guiyang 550004, China

**Keywords:** prostate cancer, Tanshinone, mTOR, Apoptosis, NF-κB

## Abstract

Prostate cancer (PCa) is the most common malignant tumor of the male urinary system in Europe and America. According to the data in the World Cancer Report 2020, the incidence rate of PCa ranks second in the prevalence of male malignant tumors and varies worldwide between regions and population groups. Although early PCa can achieve good therapeutic results after surgical treatment, due to advanced PCa, it can adapt and tolerate androgen castration-related drugs through a variety of mechanisms. For this reason, it is often difficult to achieve effective therapeutic results in the treatment of advanced PCa. Tanshinone is a new fat-soluble phenanthraquinone compound derived from Salvia miltiorrhiza that can play a therapeutic role in different cancers, including PCa. Several studies have shown that Tanshinone can target various molecular pathways of PCa, including the signal transducer and activator of transcription 3 (STAT3) pathway, androgen receptor (AR) pathway, phosphatidylinositol-3-kinase (PI3K)/protein kinase B (Akt)/mammalian target of rapamycin (mTOR) pathway, and mitogen-activated protein kinase (MAPK) pathway, which will affect the release of pro-inflammatory cytokines and affect cell proliferation, apoptosis, tumor metabolism, genomic stability, and tumor drug resistance. Thus, the occurrence and development of PCa cells are inhibited. In this review, we summarized the in vivo and in vitro evidence of Tanshinone against prostate cancer and discussed the effect of Tanshinone on nuclear factor kappa-B (NF-κB), AR, and mTOR. At the same time, we conducted a network pharmacology analysis on the four main components of Tanshinone to further screen the possible targets of Tanshinone against prostate cancer and provide ideas for future research.

## 1. Introduction

### 1.1. Current Status of PCa

PCa is the most common malignant tumor in the urinary system of men in Europe and the United States. According to the data of the World Cancer Report 2020, the number of new cases of PCa registered globally was 1,414,259, ranking second in the prevalence of male malignant tumors [1], with the highest incidence in Oceania and North America, followed by Europe. Rates in Africa and Asia are lower than in developed countries. In PCa, the incubation period is very long and is usually diagnosed in older men. At the time of diagnosis, about 90% of PCa is organ-confined or locally advanced [2,3]. Localized PCa is treated with active monitoring and local radiotherapy or resection of the prostate to achieve good treatment effects, but with the progression of the disease, advanced PCa can only be treated with surgery or chemical castration of androgen deprivation therapy (ADT), but most patients develop ADT resistance and progress to castration-resistant PCa (CRPC) in about 18 to 36 months, and once progress to CRPC is made, existing drugs and methods are often difficult to obtain effective results [4]. At the same time, the treatment of advanced PCa is still troubled by the highly toxic side effects of synthetic drugs [5]. Therefore, there is an urgent need to develop a new drug that is safe, effective, affordable, and easy to manufacture to treat androgen-independent PCa (AIPC). In recent years, natural products from fungi, plants, and animals for medical trends are emerging. Medicinal plants used for the purpose of health care in the world have increased dramatically. This is mainly because they have definite efficacy and low toxicity. In fact, more than half of the currently available drugs come from natural products [6].

### 1.2. The Basic Introduction of Tanshinone

Tanshinone is the fat-soluble component of the active ingredient of Salvia miltiorrhiza. Since the Japanese scholar Nakao first isolated Tanshinone IIA from salvia miltiorrhiza and identified its chemical structure in 1934, with the development of clinical application of salvia miltiorrhiza and the progress of extraction and separation technology in traditional Chinese medicine, the specific composition of Tanshinone has gradually become clear. So far, more than 40 Tanshinones have been isolated from salvia miltiorrhizae, among which the most important ones are Tanshinone I (TsI), Tanshinone II A (Ts II A), Dihydrotanshinone I (DHTI) and Cryptotanshinone (CYT) (Figure 1) [7,8]. The main precursor of Tanshinone biosynthesis is geraniyldiphosphate (GPP), which is derived from mevalproic acid and the 2-c-methyl-d-erythritol 4-phosphate pathway. GPP is eventually converted into Tanshinone through a series of downstream enzymes involved in various steps of catalytic biosynthesis [9,10]. Tanshinone has powerful pharmacological effects with anti-inflammatory, antioxidant stress, and anti-metabolic syndrome, and its water-soluble derivative, Tanshinone SODIUM IIA sulfonate, has been widely used in the clinical treatment of cardiovascular diseases [11]. Interestingly, more and more studies have reported the anti-tumor potential of Tanshinone, and previous studies have shown that Tanshinone can inhibit the proliferation, metastasis, and progression of various cancer cells (including PCa) by regulating transcription and growth factors, inflammatory cytokines, and intracellular signaling pathways [12,13]. Interestingly, a growing number of studies report the effects of Tanshinone on PCa cells. These studies shed light on their mechanisms of action and their potential as anti-PCa drugs. Here, we review the available evidence for Tanshinone against PCa and the molecular targets of its action.

### 1.3. Comparison of Main Components of Tanshinone

Tanshinones are uniquely characterized by the presence of 14,16-ether D rings, such as CYT and DHTI, but this heterocycle is usually further converted to furan, as in TsI and TsIIA [14,15]. Tanshinone generally consists of four rings, including naphthalene or tetrahydronaphthalene rings A and B, a normal or paraquinone or lactone ring C, and a furan or dihydrofuran ring D [15,16]. Just as A, B, C, and D rings as shown in the above figure (Figure 1). Obviously, although the molecular skeleton of the four main components of Tanshinone is similar, the groups, group positions, and double bond positions are slightly different.

Secondly, there are some differences in the pharmacokinetics of Tanshinone in vivo, but generally speaking, all Tanshinones, whether oral administration, intravenous administration, subcutaneous injection, or conventional delivery, all have the characteristics of short half-life and low bioavailability [17,18]. Interestingly, there seems to be a certain synergistic effect among the components of Tanshinone. Multiple components of Tanshinone given together can improve the bioavailability of some components, such as Tanshinone IIA and Tanshinone I, which indicates that drug interactions occur among the components of Tanshinone [19]. Tanshinone pharmacokinetics have been well summarized in previous work [20,21]. In addition, the metabolism of different types of Tanshinone in vivo is also different. It is reported that the metabolism of Tanshinones mainly depends on their saturation and substituents in their skeletons. For example, the main metabolic pathway of CYT with saturated A and D rings is dehydrogenation; DHTI with saturated D rings is mainly metabolized by D ring hydrolysis; and hydroxylation is the main metabolic pathway of TsIIA with saturated A rings [22].

In addition, there are differences among Tanshinone components in terms of antitumor pharmacological characteristics, and the conformational relationships show that the pharmacological effects of Tanshinone depend mainly on its D-ring (furan/dihydrofuran) and steroidal structure [14,23], and it has been reported that during the antitumor response, DNA molecules bind to the phenanthrene ring structure of Tanshinone, which in turn affects the synthesis of tumor DNA, while the furan ring and steroidal structure also generate free radicals, which in turn impede DNA synthesis in tumor cells [24,25]. In addition, the antioxidant effect of Tanshinone is also dependent on the D-ring, and changes in the structure of the D-ring can often affect the antioxidant capacity of Tanshinone. For example, Tanshinone containing the dihydrofuran D-ring has a stronger antioxidant capacity than those containing the furan D-ring [26], which is also reflected in the antitumor pharmacological effect. TsIIA’s tumor suppressive effect, for example, is primarily dependent on the PI3K/AKT/mTOR signaling pathway and the JNK pathway, whereas CYT is more dependent on State3-mediated anti-tumor effects and has stronger immunomodulatory effects than the other three components [27,28,29].

In addition, the antitumor potency of Tanshinone components differed among them, and the potency (either induction of apoptosis, inhibition of invasion, or inhibition of proliferation) among the components of Tanshinone was reported to be TsI > TsIIA > CYT in all three prostate cancer cell lines, whether DU145, PC3, or LNCaP [30], which may be partly due to the stronger anti-vascular activity of TsI. In addition, CYT appears to have a stronger anti-androgenic effect, but interestingly, the toxic effect of CYT on DU145 is more enhanced than that of LNCaP [31], suggesting that CYT may be more dependent on other pathways in prostate cancer.

### 1.4. Tanshinone and PCa

In a recent population-based retrospective study of 40,692 men diagnosed with PCa, the protective effect of salvia miltiorrhiza on PCa was confirmed by a 5–10% increase in survival rates among men who took salvia miltiorrhiza compared with those who did not. This protective effect is positively correlated with the dose and time of salvia miltiorrhiza use [32]. However, unfortunately, there is only one study on clinical evidence of salvia miltiorrhiza and PCa at present, and in this study, the experimenters also did not further verify whether the intake of salvia miltiorrhiza could inhibit the occurrence of castration-resistant PCa, the occurrence of biochemical recurrence of PCa after radical prostatectomy, and the increase in the aggressivity of PCa.

However, there are many in vitro studies on Tanshinone and PCa cell lines. Human PCa cells: LNCaP, PC3, DU145, and 22RV1, which are commonly used as cell models for PCa in vitro experiments, have different characteristics, respectively. For example, LNCaP has functional ARs, so it is sensitive to androgen and secretes prostate-specific antigen (PSA). 22RV1 cells were androgen-sensitive but not androgen-dependent and showed low aggressiveness.PC3 cells do not depend on androgens, are highly invasive, and have a strong potential for metastasis.DU145 cells are also androgen-independent cells with strong proliferation ability but only moderate metastasis ability [33]. In these in vitro experiments, the researchers pretreated PCa cell lines with varying biological characteristics with a specific concentration of Tanshinone and then used various modern molecular biological methods to identify proteins involved in the cell cycle, apoptosis, growth, and metastasis. Finally, they confirmed that Tanshinone in vitro by adjusting the related protein expression and signaling pathway in PCa cells induced the stagnation of the cell cycle and apoptosis, inhibiting metastasis and invasion of tumor cells (Figure 2). We put the current relevant Tanshinone in vitro effect on Pca-related research summarized in Table 1.

Similarly, in vivo experiments with Tanshinone against PCa are also under way. In the experiments, the researchers transplanted human PCa cells (LNCaP, PC3, and 22Rv1) subcutaneously or in situ into immunodeficient mice that did not reject human cells to establish animal models. Finally, it was confirmed that Tanshinone could effectively inhibit the growth of tumors in vivo, whether injected orally or subcutaneously/intraperitoneally. It is worth noting that in the current animal experiment, it has not been observed that the activity of Tanshinone on other organs and tissues, except tumor tissues, has toxic effects. Table 2 summarizes the current state of Tanshinone and PCa in vivo research.

## 2. Tanshinone as a Potential Anti-Cancer Agent for PCa

### 2.1. Tanshinone-Induced Stagnation of the PCa Cell Cycle

A normal cell cycle is essential for homeostasis and normal development of an organism, and dysregulation of this system often leads to uncontrolled cell proliferation, leading to the occurrence of tumors [61]. The progress of the cell cycle depends on the regulation of various cyclins, cell cycle-dependent protein kinases (CDK), and CDK inhibitors. In mammals, the normal progress of the cell cycle is carefully edited with the expression of different cyclin proteins in characteristic cell cycle stages as the center of the molecular mechanism, and they are assembled with specific CDKs to promote the phosphorylation of the retinoblastoma tumor suppressor (Rb) so as to promote the cell cycle to the next stage [62,63]. Therefore, in order to maintain their own biological characteristics (uncontrolled proliferation), tumor cells induce protein expression of cell cycle progression and the genes encoding the cell cycle inhibitors are missing, or there is inactivation of cell cycle regulators such as Rb and P53 [64,65]. Interestingly, current publications suggest that Tanshinone can significantly inhibit PCa proliferation by inducing PCa cell cycle arrest. The mechanism is that Tanshinone can significantly reduce cyclinD1, A, and E (cyclinD1, A, and E) in PCa cells [34,51,54,56], which seems to be partly due to Tanshinone’s ability to directly inhibit the phosphorylation of the pro-proliferative signaling pathway STAT3Tyr705 in PCa cells [51]. Tanshinone also has a significant ability to relate to CDK family-relative proteins in PCa cells. Previous studies have shown that Tanshinone can not only inhibit the expression of CDK in PCa cells but also increase the activation and expression of CDK inhibition (P21, P27, p16) by promoting the phosphorylation of Ser15 residues of p53 [34,59]. It can also significantly inhibit the hyperphosphorylation of Rb in PCa cells [54], which seems to be caused to some extent by Tanshinone increasing ROS in PCa cells [35,66]. In addition, the normal of the cell cycle also depends on other factors of the adjustment. These factors are not only for genomic stability and integrity maintenance, which is very important, but they also participate in the spindle and maintain normal mitosis of the structure of the adjustment. It is interesting to note that, according to the existing literature, Tanshinone and its derivatives in PCa cells also show the effect on the regulation of these factors. including: stasis and DNA damage-inducing protein (GADD45A), polo-like kinase 1 (PLK1), and checkpoint-related protein [54].

### 2.2. Tanshinone-Induced Apoptosis of PCa Cells

According to reports, tumor progression frequently means apoptosis and proliferation imbalances, which are related to apoptosis signaling molecules and proteins that scientists are concerned about [67,68]. Interestingly, previous studies have shown that Tanshinone induces apoptosis in PCa cells by altering the balance between the expression of pro-apoptotic and anti-apoptotic proteins in the b-cell lymphoma-2 (bcl-2) family [34,35,38,50,51,55,56,57,58]. Secondly, caspase containing cysteine is a key enzyme for performing apoptosis, and caspase-3 is a key executor of apoptosis in mammalian cells [68]. Studies have shown that Tanshinone can significantly induce its expression and phosphorylation in PCa. It also significantly increases the activation and expression of caspase-8 or-9 upstream promoters [34,35,38,50,51,55,56,57,58]. The regulation of these proteins by Tanshinone is partly due to the inhibition of the pi3k/akt pathway and MAPK pathway and the expression of hypoxia stress factor 1 (HIF-1) in PCa cells [38,45]. In fact, Tanshinone also has a significant effect on the regulation of ROS and LC3II in PCa cells for the process [36]. In addition, Tanshinone induces apoptosis of PCa cells by other mechanisms. As the report goes, Tanshinone activates mitochondrial-dependent apoptosis of PCa cells by inhibiting the expression of mitochondrial protective Bcl-2 family protein Mcl-1 by inducing the cleavage of ADP ribose polymerase (PARP), promoting the release of cytochrome c from the mitochondria to the cytoplasm and reducing mitochondrial membrane potential. which seems to be partly due to the inhibition of pik3/Akt in PCa cells [38]. Similarly, Tanshinone derivatives also have the same pharmacological effects. In the experiments conducted by Wang et al., they found that Tanshinone derivatives induce PCa cell apoptosis by regulating p53, ERK1, BAX, P38, Bcl-2, caspase-8, cleavedcaspase-8, and PARP1, and significantly affect the phosphorylation of ERK1 and P38 in P38 [54]. Second, Tanshinone has been shown to induce ER stress pathway apoptosis in PCa cells by increasing the expression of key proteins of ER stress pathway apoptosis, such as glucose regulatory protein 78 (BIP/GRP78), ER stress sensor (IRE1-) and its downstream target CAAT/enhancer binding protein homologous protein/growth arrest and DNA-damage-inducing gene 153 (gadd153/cho) [48,53]. In addition, Tanshinone has been reported to up-regulate Fas sensitivity of PCa cells and promote Fas (APO1/CD95) mediated apoptosis of PCa cells by inhibiting phosphorylation of Janus kinase (JAK) and p38MAPK [45]. In addition, Tanshinone has been shown to be an effective sensitizer of tumor necrosis factor-associated apoptosis-inducing ligand (TRAIL) to enhance TRAIL-mediated apoptosis, which seems to be related to Tanshinone activation of mir135A-3p mediated up-regulation of death receptor 5 (DR5) in PCa cells [52].

### 2.3. Tanshinone-Induced Motility Inhibition of PCa Cells

Metastasis of tumor cells is often one of the main causes of death in patients. Previous studies have shown that prostate cancer can metastasize through a variety of mechanisms. Among these are the secretion of matrix metalloproteinases (MMPs) to dissolve the extracellular matrix, the promotion of angiogenesis, the recruitment of some cytokines and chemokines, and so on [69,70], which pose significant challenges to clinicians’ diagnosis and treatment. Fortunately, existing literature shows that Tanshinone can act as an effective inhibitor of the metastasis and invasion of PCa cells. In the experiment of WuCY et al., Tanshinone inhibits the translocation of phosphorylated STAT3 and protein expression of P-STAT3 and Skp2 in PCa cells in a dose-dependent manner, resulting in inhibition of the translation and transcription of RhoA and SNAI1 genes in PCa cells, which results in reduced invasiveness of PCa cells [35]. Interestingly, inhibition of this pathway also has a significant impact on the ability of PCa cells to recruit macrophages. Inhibition of this pathway will lead to reduced secretion of related cytokines in PCa cells [32], including CCL2, CCL5, interleukin-1 receptor antagonists (il-1ra) and intercellular adhesion molecule-1, which are significantly associated not only with PCa metastasis but also with the ability to recruit macrophages of Pca [71,72]. In addition, literature has shown that Tanshinone has a significant pharmacological effect of inhibiting angiogenesis in PCa cells, which seems to be related to the inhibition of hif-1 expression induction of vascular endothelial growth factor and its receptor in PCa cells by Tanshinone intervention [30,31]. Similarly, Tanshinone derivatives also have significant inhibitory effects on the metastasis and invasion of PCa cells. In the experiment of Wang et al., they found that Tanshinone derivatives can reduce the expression of protein molecules related to metastasis and invasion in PCa cells, including MMP-1, MMP-9, and VEGF-1 [54].

### 2.4. Tanshinone Maintains Gene Stability of PCa Cells

Tanshinone also regulates the expression of epigenetic-modification-related genes in PCa cells. Among 84 epigenetic-modification-related genes in PCa cells treated with Tanshinone (mainly TsI), the expression of 32 genes was down-regulated [30], including AuroraA kinase, DNA methyltransferase, histone acetyltransferase, histone deacetylase, Lysine (K)-specific demethylase, and protein arginine methyltransferase. The PCa cells treated with CYT and TsIIA only had the expression of AuroraA kinase inhibited. However, previous studies have shown that epigenetic disorders such as histone modification and DNA methylation contribute to the initiation and progression of PCa. AuroraA kinase is a key mitotic regulator required to maintain chromosome stability [73], and its overexpression often indicates a higher degree of malignancy in tumors. In addition, Tanshinone has been reported to be significantly associated with the down-regulation of astrocyte elevating gene 1 (AEG-1) in PCa cells [31], which is involved in the regulation of multiple signaling pathways in cancer cells, including PI3K/Akt, NF-κB, Wnt/-catenin, and MAPK. They synergistically promote the oncogenic and metastatic potential of transformed cells [74]. In addition, Tanshinone can reduce the transcription and translation of topoisomerase 2A in PCa cells [52], which is critical for genomic stability and replication. Interestingly, Tanshinone has been reported to regulate the activity and elongation of the RNA-binding protein HuR and telomerase. This coordinates mRNA stabilization and translation [75,76,77,78], which is critical for inflammation and tumor progression.

In fact, the antioxidant capacity of Tanshinone also plays a significant role in maintaining the genetic stability of prostate cancer cells. Although this has not been confirmed, studies have shown that oxidative stress can often damage DNA molecules, leading to the occurrence of tumors [79]. However, the specific antioxidant activity of Tanshinone is that it can effectively inhibit the interaction between DNA and intracellular lipid peroxidation products, thereby stabilizing DNA molecules [80]. However, the current research does not pay attention to the influence of Tanshinone on the factors and pathways related to oxidative stress in prostate cancer cells. Secondly, the special skeleton structure of Tanshinone also has an effect on the stability of DNA. It is reported that the Tanshinone D ring can bind to the small grooves in DNA molecules, thereby stabilizing DNA molecules [23,25]. In addition, the regulatory effect of Tanshinone on microRNAs in prostate cancer cells seems to be involved in this process. After all, microRNAs are closely related to gene expression and synthesis, and mutations of microRNAs often lead to cancer [81,82]. Unfortunately, although Tanshinone has been proven to be able to regulate microRNAs in other cancer cells [27,83], the current study has not confirmed the regulatory effect of Tanshinone on microRNAs in prostate cancer cells. Based on the fact that microRNA not only plays a key role in the process of gene expression and synthesis but also has significant significance in maintaining the malignant behavior of cancer cells, future research should also pay attention to the regulatory effect of Tanshinone on microRNA in prostate cancer cells. In conclusion, the current research is limited. Although laboratory data have confirmed that Tanshinone has the ability to stabilize the genetic stability of prostate cancer cells, there is no exact evidence and the corresponding mechanism is not clear.

### 2.5. Tanshinone Reverses Multidrug Resistance in PCa

Tanshinone also has a regulatory effect on multidrug resistance (MDR) of tumor cells, and MDR is the main factor for clinical chemotherapy failure. Although chemotherapy is not recommended as a first-line treatment for PCa in the early stages of the disease, it is often recommended as a first-line treatment for PCa in the advanced stages [84,85]. Existing literature shows that Tanshinone can adjust the genes associated with MDR (P glycoprotein, topoisomerase, lung resistance protein expression) to reverse the drug resistance of tumor cells to chemotherapy drugs [86]. Although this has not been confirmed in PCa, it is interesting to note that after Tanshinone pretreatment, the toxic effects of cisplatin and azithromycin on PCa increased [36,60]. In addition, tumor stem cells are also considered to be one of the key factors in the occurrence of MDR [87]. It is noteworthy that Tanshinone has also shown a regulatory effect on tumor-initiating cells in PCa. In an experiment, YingZhang et al. found that Tanshinone can regulate the expression of prostatic cancer cell globogenesis and stem cell genes (Nanog, OCT4, SOX2, -catenin, CXCR4) and simultaneously change the cell proliferation, cell cycle state, migration, and colony formation of prostate tumor-initiating cells (CD44 + CD24-population) [57]. This could be because Tanshinone significantly reduced Bmi1 gene expression and protein levels (key regulators of stem cell self-renewal and malignant transformation) [88].

Tanshinone can prevent the occurrence of MDR in prostate cancer cells by down-regulating the genes related to the occurrence of MDR and inhibiting the tumor-initiating cells closely related to the occurrence of MDR. It is worth noting that Tanshinone can also greatly improve the adverse reactions caused by radiotherapy and chemotherapy [89]. Research shows that Tanshinone can significantly improve the organ nerve loss caused by radiotherapy and chemotherapy [89,90]. Therefore, based on the current research, we believe that Tanshinone combined with radiotherapy and chemotherapy in the treatment of drug-resistant prostate cancer will become a possibility. However, the current research is limited, and the specific mechanism of Tanshinone enhancing the toxic reaction of chemotherapy drugs to prostate cancer is still unclear, so further research is needed.

### 2.6. Tanshinone Changes the Metabolic Process of PCa

Tanshinone has also been reported to regulate the metabolic process of PCa. Existing publications show that the expression of some genes related to steroid and cholesterol biosynthesis in PCa cells treated with Tanshinone (mainly Tanshinone IIA) is significantly reduced [91], including methylsterol monooxygenase 1 (MSMO1/SC4MOL), squalene monooxygenase (SQLE), ATP-binding box subfamily G member 1 (ABCG1) and preprotein invertase subtilin kexin9 (PCSK9s). Although this has not been proven to be associated with tanshoneinhibition of PCa, it is worth mentioning that Tanshinontong changes the way tumor cell metabolism works, including down-regulating the enzymes involved in glucose uptake and metabolism in cancer cells (especially those related to glycolysis), affecting the energy metabolism of tumor cells, and then inducing apoptosis of cancer cells [92,93].

It is reported that in the process of progression, tumors often change their metabolism to quickly obtain energy or synthesize substances they need [94]. This process is critical for prostate cancer. Previous studies have shown that advanced prostate cancer may be more dependent on lipid metabolism. The expression of genes related to lipid synthesis in advanced prostate cancer cells is significantly higher than that in early prostate cancer cells [95,96]. Prostate cancer increases the synthesis of its own lipids in this way, thereby providing more abundant raw materials for androgen synthesis so as to tolerate the low level of androgen brought on by castration treatment [96,97]. Therefore, regulating the metabolism of prostate cancer has significant significance for the treatment of prostate cancer. As evidenced based on appeal, the metabolic rearrangement of Tanshinone on prostate cancer cells should also be paid attention to in future research. The molecular mechanism involved should also be further explored.

## 3. Molecular Targets of Tanshinone Action

### 3.1. Tanshinone and NF-κB

NF-κB is a multipotent transcription factor responsible for regulating cell signaling and various biological processes, such as immune response, inflammation, cell transformation, cell proliferation, angiogenesis, cancer invasion, and metastasis [98]. This factor can be activated by a variety of signals, including growth factors, protein kinases, oxidative stress inducers, mitogens, pro-inflammatory cytokines, and chemokines (TNF-, Il-1, Il-8, Il-6, CXCL12), and environmental stress factors [98,99]. In addition, NF-κB has been proven to be significantly related to the occurrence and progression of PCa. During the development of PCa, NF-κB can promote the survival, invasion, angiogenesis, metastasis, and chemical resistance of cancer cells by inducing pro-survival genes (such as bcl-2 and bcl-xl), pro-inflammatory cytokines, and vascular growth factors [100,101]. Second, IκB kinase -α (IKKα) and IκB kinase -β (IKKβ) signaling molecules upstream of the NF-κB signaling pathway can directly affect AR activity via phosphorylation and induce the expression of the constitutive active AR splicing form (AR-v7), which is one of the key factors in PCa castration resistance and drug resistance [102,103]. Interestingly, available laboratory data have long shown that Tanshinone inhibits cancer cell growth and progression by inhibiting the overactivation of NF-κB in cancer cells [104,105], although this has not yet been demonstrated in PCa cells. Tanshinone has been shown to inhibit the secretion of NF-κB activating factors such as interleukins, vascular cell adhesion molecule-1 (VCMM-1), and intercellular adhesion molecule-1 (ICAM-1) as well as monocyte chemoattractant protein 1 (MCP-1) and tumor necrosis factor (TNF-α). It can also directly inhibit the phosphorylation of upstream molecules by IB kinase-IKK and IKK and simultaneously induce degradation of IKK and IKK. It can also inhibit the NF-κB pathway by decreasing the expression levels of toll-like receptor (TLR), myeloid differentiation factor 88 (MyD88), transferrin 6 (TRF6), and other proteins involved in the NF-κB signaling pathway and directly inhibiting the activation and expression of NF-κB [106,107,108]. It is noteworthy that Tanshinone can inhibit the activity of COX2 and play a significant inhibitory role in the transcription and expression of COX2 by acting similarly to non-steroidal anti-inflammatory drugs [108,109,110], while the prostaglandins generated by COX2-mediated production not only have a direct nourishing effect on PCa cells [111]. It also acts as an effective activator of NF-κB

Inflammation is essential for the occurrence and development of all kinds of cancers, including prostate cancer. The signal pathway most closely associated with the inflammatory response is NF-κB. It not only promotes the onset and progression of prostate cancer but also has a clear crosstalk relationship with androgen receptor signals. In fact, it has crosstalk with most signal molecules in the cell. Therefore, based on the current laboratory data, targeting NF-κB is probably a promising treatment [112]. Tanshinone can inhibit the occurrence and progression of prostate cancer by virtue of its excellent ability to regulate inflammation and NF-κB. Unfortunately, current studies have not confirmed the effects of Tanshinone on inflammatory cytokines and the NF-κB pathway in prostate cancer cells. Therefore, future studies should pay attention to the regulation of inflammation-related molecules and signaling pathways in prostate cancer.

### 3.2. Tanshinone and AR

Androgen is very important for PCa. It is reported that testosterone, after binding to hormone ligands, is transferred to the nucleus and combined with androgen response elements (ARE) located in the promoter region of genes involved in cell proliferation and escape from apoptosis to promote the growth and development of PCa cells. It can also promote the growth and development of PCa by activating other extracellular signaling pathways that crosstalk with androgen signaling pathways [113,114]. Androgen deprivation therapy (ADT) is widely used as a first-line treatment method for metastatic androgen-dependent PCa. However, accept the ADT after treatment of PCa, often in about two years into a more aggressive PCa (CRPC) [4]. Previous views that CRPC does not depend on the nourishment effect of androgens; however, recent laboratory evidence suggests that CRPC still relies on androgen to nourish them either by synthesizing more bioactive androgen through themselves and the adrenal cortex or by increasing the expression of ARs and inducing them to mutate into a more active form to tolerate low androgen levels induced by castration therapy [115,116,117]. Therefore, inhibition of androgen nourishing effects is of great significance for the treatment of PCa.

Interestingly, Tanshinone inhibits androgen signaling through a variety of mechanisms. Cryptotanshinone has been reported to have a chemical structure similar to dihydrotestosterone (DHT), the most effective androgen for AR activation, and can bind to AR receptors as a competitive antagonist of DHT, thereby inhibiting DHT-mediated AR trans-activation [46]. Tanshinone also has a significant regulatory effect on the expression and activity of AR. According to current published reports, Tanshinone can not only directly inhibit the expression of AR [34,40], but also increase the monomethyl and dimethylation of lysine 9 (H3K9) of histone H3 by targeting the lysine-specific demethylase 1 (LSD1) complex. In addition, AR activity and expression can be inhibited by inhibiting AR N and C terminal dimerization and the formation of an AR-regulator complex [46,55], and reducing the availability of AR by overexpressing heat shock protein (Hsp90) can be changed [30], and it also has a regulatory effect on ARE [46]. Tanshinone, by regulating ARE, can inhibit the transcriptional regulation of AR signaling to its target genes. In addition, Tanshinone also regulates AR receptor mutations. In a study by Liu et al., Tanshinone IIA and its derivatives not only significantly inhibited AR expression but also acted as effective inhibitors of AR receptor mutations, thereby inhibiting AR receptor mutations [41]. Furthermore, it also has a certain regulatory effect on androgen secretion. Previous studies have shown that Tanshinone can not only regulate the expression of key enzymes related to androgen biosynthesis but also regulate extracellular signal-regulated kinase (ERK)/C-FOS/17, 20-lyase (CYP17), leading to androgen biosynthesis levels being down [118,119]. However, it is worth noting that, in addition to having similar pharmacological effects, Tanshinone has other advantages over existing anti-male drugs. As reported, Tanshinone derivatives can significantly inhibit AR trans-activation mediated by 17-estradiol (E2) and androgen-5--Δene-3, 7-diol (Δ5-androstenediol or Adiol) [55]. E2 and Adiol are natural hormones in PCa cells. Because current adt-related drugs do not target these two natural hormones, and because they have the characteristics of converting to testosterone and even acting as effective activators of the AR signaling pathway, they are significantly associated with drug resistance and castration resistance in PCa [120,121]. Secondly, in a clinical trial, LinTH et al. found that although the use of enzaluamide and bicaluamide could lead to the reduction of the primary tumor and PSA, the invasion of PCa cells was significantly enhanced, while the use of Tanshinone could not only achieve the reduction of the primary tumor and PSA but also inhibit PCa cell invasion to a certain extent [53]. Similarly, for other steroid receptors present in PCa, Tanshinone has been reported to inhibit prostatic stromal and epithelial proliferation by down-regulating estrogen receptorα (ERα) [122]. Interestingly, ER is limited to basal cells and stromal cells of the prostate epithelium [123]. It is associated with mitogen-activated protein kinase (MAPK) activity and maintenance of the phosphoinositol 3-kinase (PI3K) signaling pathway in PCa [124].

In short, the occurrence and progression of prostate cancer are inextricably linked to the nourishing effect of androgens. Even if it is CCRP, inhibiting the nourishing effect of androgen in prostate cancer is still the primary purpose of endocrine therapy for prostate cancer. Therefore, as previously described, Tanshinone has a significant inhibitory effect not only on androgen biosynthesis but also on the expression and synthesis of various proteins and genes related to the androgen pathway. Tanshinone has been shown to be more effective than some anti-androgenic drugs because, when compared to these anti-androgenic drugs, Tanshinone not only achieves the same effect but also inhibits prostate cancer cell metastasis. In fact, compared with traditional antiandrogenic drugs, the inhibition of the androgen-mediated signaling pathway by Tanshinone is multi-pathway and multi-target. These data once again confirm the possibility of Tanshinone as an anti-prostate drug.

### 3.3. Tanshinone and mTOR

mTOR is a conserved and universally expressed serine-threonine kinase. In mammals, it is usually assembled with Raptor, rictor, lst8, and msin1 to form two catalytic subunits of different protein complexes (mTORC1 sensitive to rapamycin and mTORC2 insensitive to rapamycin) and participates in a variety of signal pathways in vivo to regulate cell proliferation, autophagy, and apoptosis. Among them, mTORC1 is the most important. mTORC1 not only consists of three core components: mTOR, regulatory-associated protein of mTOR (raptor), and mammalian lethal with SEC13 protein 8 (mlst8), but also contains two inhibitory subunits: 40 kDa proline-rich Akt substrate (PRAS40) and regulatory-associated protein of mTOR (deptor) [125,126]. Here, we mainly review the regulatory effect of Tanshinone on mTORC1.

In terms of assembly, Tanshinone can not only directly inhibit the expression and phosphorylation of mTOR so that it interferes with the assembly of the mTORC1 subunit [23], but also inhibit the promotion of mTORC1 assembled catalyst and stabilizer (heat shock protein 90) [30,127,128]. In terms of activity, the activity of mTORC1 is regulated by growth factors, cell energy, stress, and nucleotides, and the lysosome is the main site of its activation [126,129]. In fact, the activity of mTORC1 is mainly related to the tuberous sclerosis complex (TSC), which can inactivate Ras homolog enriched in the brain (rheb) (an important activator of mTORC1), thereby inhibiting mTORC1 [130]. Interestingly, the existing literature shows that Tanshinone can inhibit not only the expression of growth factors and their receptors [131] but also the expression of rheb [132]. In addition, the activity of mTORC1 is also regulated by the energy sensor AMP-dependent kinase (AMPK). AMPK can promote the inhibition of rheb by TSC by stimulating the gap activity of TSC, resulting in the down-regulation of mTORC1 [133,134], which means that activating AMPK in cancer cells can affect mTORC1 to a certain extent. Interestingly, Tanshinone has been shown to inhibit the mTORC1-mediated signal pathway by activating the AMPK-TSC2 axis [135]. Sestrins negatively regulate mTORC1 signaling through GATOR2/Rag and its components, SESN1 and SESN2, can directly bind to complex and AMPK, resulting in AMPK activation and autophosphorylation in a p53-dependent manner. This stimulates AMPK-mediated TSC2 phosphorylation to negatively regulate mTORC1 signaling [136,137]. Previous literature shows that Tanshinone induces the expression of BECN1 and SESN2 proteins in a dose-dependent manner and then induces autophagy in osteosarcoma cells [138]. It also regulates the upstream gene p53 of the SESN2 protein. Secondly, the down-regulation of pyruvate dehydrogenase kinase 4 (PDK4) has also been reported to lead to the inactivation of mTORC1 [139], and Tanshinone is known as an inhibitor of PDK4, which means that the inhibition of mTORC1 activity by Tanshinone may be partly due to the inhibition of PDK4 [140]. In addition, mitogen-activated protein kinase (MAPK) downstream of the growth factor receptor can also up-regulate mTORC1 activity. MAPK-related signal molecules can promote the activation of mTORC1 through the phosphorylation of Raptor. For example, MEK1/2 can not only phosphorylate Raptor but also promote Raptor phosphorylation through ERK1/2 and P90 ribosomal S6 kinase (RSK1/2). This increases mTORC1 activity [141,142]. Secondly, RAS and RAF kinases, key molecules in the MAPK-mediated signal pathway, also regulate the activation of its upstream signal molecule, PI3K [143,144]. In fact, MAPK and mTOR-mediated signaling pathways have mutual crosstalk in many aspects. They not only receive the activation of various growth factors but also have the same downstream molecules, such as Src kinase, FOXO (forkhead box o), c-myc transcription factor, and various metabolism-related enzymes [143]. This means that inhibition of this pathway can inhibit the activation of mTORC1 and its related pathways to a certain extent. Tanshinone not only directly inhibits the expression and phosphorylation of MAPK but also inhibits the activation and expression of related molecules of its mediated signal pathway, including ERK, JNK, p38 MAPK, Ras, etc. [45,54,145,146]. It can also indirectly affect MAPK by up-regulating ROS in cancer cells [66].

PI3K is a large class of lipid kinases and one of the most important upstream molecules of mTORC1-mediated related pathways. At present, it is mainly divided into three categories: class I (subdivided into classes IA and IB), class II, and class III, of which class I is the most important [142], and class IA PI3K is a heterodimer that consists of a catalytic subunit (P110α, p110β or P110δ) and a regulatory subunit (p85α/p55α/p50α, p85β, or p55γ). It is activated by various growth factors. After being activated, class IA PI3Ks synthesize lipid secondary messenger phosphatidylinositol 3,4,5 triphosphate (PIP3) from phosphatidylinositol 4,5 diphosphate (PIP2), which recruits protein kinase Akt to the plasma membrane, where it is activated by 3-phosphoinositide dependent kinase 1 (PDK1), Subsequently, activated Akt phosphorylates TSC2 and inhibits the TSC complex, eventually leading to the activation of mTORC1 [142,147]. However, as described above, Tanshinone can inhibit the expression of various growth factors and their receptors. In fact, Tanshinone can weaken or even inhibit the induction of PI3K protein expression and phosphorylation by these factors, which is partly because Tanshinone directly inhibits PI3K protein expression and phosphorylation [148,149,150]. Secondly, Tanshinone also showed the pharmacological action of inhibiting the phosphorylation and expression of catalytic subunit P110α/γ and subunit p85 (down-regulating the expression of its gene) [38,151,152,153]. In addition, Tanshinone has the pharmacological effect of promoting the expression of phosphatase and tensin homologue (PTEN), a negative regulatory gene of PI3K [133,154,155], which negatively regulates PI3K Akt mTOR signal transduction by transforming PIP3 back to PIP2 [156] and the deletion of this gene usually occurs with the progression of prostate cancer [157,158]. In addition, the expression and phosphorylation of PDK1 and Akt downstream of PI3K were also inhibited by Tanshinone [148,149,150,159].

The two most important molecules in mTORC1-mediated cell activity are P70S6 kinase 1 (S6K1) and eIF4E binding protein (4E-BP) 1 [160]. The phosphorylation of S6K1 by mTORC1 leads to increased protein and nucleotide synthesis [125,126,160]. 4ebp is a negative regulator of 5′cap dependent mRNA translation, and mTORC1 induces the separation of 4E-BP1 from eIF4E so as to reduce its inhibition of protein synthesis [125,126,160]. By targeting these two molecules, mTORC1 synthesizes proteins required for cell growth, cell cycle process, and cell metabolism, and then induces tumor growth and progression. Tanshinone also has a regulatory effect on these two molecules. As shown in previous literature, Tanshinone can not only prevent the binding of S6K1 to mTORC1 [161] but also significantly inhibit the expression and phosphorylation of S6K1 and 4E-BP1 [159,162]. It is worth mentioning that Tanshinone also regulates the key downstream molecule PKC of mtorc2 [128]. Protein kinase C (PKC) is considered to be the main downstream molecule of mtorc2. Mtorc2 completes various cell activities by targeting the special type of this molecule [126].

This laboratory evidence shows that Tanshinone can inhibit mTOR-mediated tumor malignant behavior in prostate cancer cells by regulating the expression and phosphorylation of upstream and downstream protein molecules of the mTOR-mediated signaling pathway or by regulating other molecular pathways with obvious crosstalk with mTOR. It is worth noting that although Tanshinone is similar to rapamycin in structure, the regulation of mTOR does not directly inhibit the synthesis of mTOR like rapamycin, which means that Tanshinone has fewer side effects on normal tissue and is safer [163,164]. In short, Tanshinone has a significant inhibitory effect on prostate cancer. its pharmacological mechanism. As previously described, it depends on the regulation of the mTOR-mediated signaling pathway, but the current research is still limited, The regulatory mechanism of some Tanshinones on mTOR has only been confirmed in other tumor cells. Therefore, a large number of studies are still needed to further clarify the effect of Tanshinone on the mTOR-mediated signaling pathway in prostate cancer cells. Furthermore, State3 [27] is another key molecule of Tanshinone that exerts an anti-tumor effect. Although it has been confirmed to be involved in the inhibitory effect of Tanshinone on prostate cancer, the current research is very lacking and should be given enough attention in the future. We plotted the relevant regulation of Tanshinone on the mTOR pathway in Figure 3

## 4. Dilemma of Clinical Application of Tanshinone

Tanshinone, as a secondary metabolite, accumulates mainly in the roots of Salvia miltiorrhiza, but in very low yields. Currently, Tanshinone relies on traditional chemical isolation and purification from Salvia divinorum roots [8]. However, traditional methods are characterized by low efficiency, high energy consumption, and unfriendliness to the environment and plant resources [8,165], and secondly, due to the increasing demand for Tanshinone in the market, wild salvia has been over-harvested and the resources are on the verge of extinction [8,165]. Although artificial domestication of salvia has been cultivated since the 1970s, due to the low yield of secondary metabolites and the long growth period of cultivated plants, the production of Tanshinone from cultivated salvia cannot meet the rapidly growing market demand [17]. Therefore, both the reform of the purification process of Tanshinone and the improvement of yield through modern biotechnology have received great attention. Various in vitro culture systems of Salvia divinorum, including suspension cells, guard tissues, adventitious roots, hairy roots, and new techniques such as the use of endophytic fungi and transgenic plants, have been reported to significantly increase the yield of Tanshinone [18,166]. However, these efforts are still insufficient for the increased demand for Tanshinone in the market, and therefore a new technique with a higher yield is urgently needed to provide the supply.

In recent years, the discovery of key genes for the biosynthesis of pharmaceutical active ingredients and the use of synthetic biology strategies to design and modify microbial strains to produce natural products are considered to be promising resource acquisition methods [167]. It is reported that at present, with Saccharomyces cerevisiae as the chassis cell, a high-yield engineering strain of miltiradiene, an important intermediate of Tanshinone, with a yield of up to 488 mg/L has been constructed through functional module design, and cloned the modifying enzyme gene CYP76AH1 of the Tanshinone biosynthesis pathway, which successfully converted miltiradiene into ferruginol [17,168]. However, due to the limited understanding of the transcription genes of Salvia miltiorrhiza, the further elucidation of Tanshinone biosynthesis has been hindered [167,169]. It is worth mentioning that the formation of the furan ring, which has puzzled scientists for a long time, has been gradually decoded [170]. However, there is still a long way to go. It is still necessary to pay more attention to and explore genes and enzymes related to Tanshinone biosynthesis in the future.

Second, like most natural drugs, Tanshinone also has the characteristics of low water solubility, poor stability, large first-pass elimination, and low bioavailability [17,171], which greatly limits the clinical application of Tanshinone. It is reported that when Cryptotanshinone is administered at a body weight of 100 mg/kg (mg/kg), the bioavailability of oral and intraperitoneal injection in rats is 2.1% and 10.6%, respectively. In fact, most of the studies on Tanshinone have pointed out the nanomolar to lower micromolar after oral administration. The maximum/peak concentration (Cmax) value within the range [17,18] greatly limits the clinical application of Tanshinone. Although the currently prepared new formulations of Tanshinone for injection, such as microemulsion, microspheres, solid dispersion, liposomes, and nanoparticles, can significantly improve the bioavailability of Tanshinone, the complexity of the process, high cost, and low tissue specificity still limit the clinical application of these formulations [166,171]. However, it is worth noting that in previous studies, researchers greatly enhanced the specificity of Tanshinone nanoparticles for prostate cancer tissue by combining them with prostate-specific membrane antigen [60], which provides a way for the development of Tanshinone anti-prostate cancer-related nanoparticles in the future. However, the clinical trials of Tanshinone nanoparticles are still lacking, and the clinical application of Tanshinone nanoparticles is still making little progress. In addition, Tanshinone derivatives by changing the Tanshinone skeleton group also seem to be a promising solution, but in most cases, the anti-tumor effect of Tanshinone derivatives will be reduced or even lost, for example, sodium Tanshinone sulfonate [172,173]. Although previous studies have found some Tanshinone derivatives with significantly increased bioavailability and anti-tumor efficacy [42,54,55], due to the lack of in vivo experiments and related toxicological experiments, the safety of these Tanshinone derivatives is also a worrying problem and is also limited by the lack of raw materials. Therefore, future research should not only explore new Tanshinone derivatives but also focus on the safety of Tanshinone derivatives.

In addition, the safety of Tanshinone is also an aspect of concern for scientists. Although the experiments of Wang et al. confirmed that in both acute and subchronic toxicity studies, no abnormalities of other organs were observed in Sprague Dawley rats treated with Tanshinone injection, except that it caused focal inflammation at the injection site [174], which is consistent with the conclusions drawn from the in vivo experiments of Tanshinone and prostate cancer. In fact, limited studies pointed out that high concentrations of TsIIA and CYT showed serious growth inhibition, developmental malformation, and cardiotoxicity to zebrafish embryos [175,176]. Similarly, high concentrations of TsIIA were also toxic to normal human endothelial cells. It was reported that high concentrations of (25 uM) TsIIA could kill endothelial cells within 24 h [177]. Therefore, future research should focus on the toxic reaction of Tanshinone to normal tissues to confirm the safety and stability of Tanshinone, which is crucial for the development of clinical drugs.

## 5. Conclusions and Prospects

It is well understood that blocking the products of a single signal pathway or gene is frequently insufficient to prevent or treat malignant tumors. Tanshinone is expected to be a candidate drug for the treatment of prostate cancer because it can regulate the proliferation, survival, migration, and metabolism of prostate cancer through multiple targets, links, and pathways. Although the current experiments have confirmed the significant anti-prostate cancer effect of Tanshinone, there are still some problems with the current research. First, the current research on the anti-prostate cancer effect of Tanshinone is still limited, and most of the studies remain to verify the expression of proteins and genes related to apoptosis, cell cycle, and invasion in prostate cancer cells. In order to further explore the specific molecular mechanism that causes these changes, more research is still needed to explore the specific mechanism of Tanshinone against prostate cancer in the future. In addition, as described above, TsI and DHT have stronger anti-prostate cancer effects. However, current research is mostly focused on TsIIA and CYT. TsI and TsIIA should be given sufficient attention in the future. In addition, Tanshinone can significantly regulate the tumor immune microenvironment, which is more important for prostate cancer with poor immunotherapeutic efficacy. In the future, we should also pay attention to the effect of Tanshinone on the immune microenvironment of prostate cancer.

Although Tanshinone has been widely reported to be beneficial to health, its clinical application is still subject to many restrictions. First, as a secondary metabolite, Tanshinone mainly comes from the root of Salvia miltiorrhiza. Due to the characteristics of the long growth cycle and low yield of cultivated plants, the supply of Tanshinone is difficult to meet the market demand, although the current research on Tanshinone biosynthesis has made good progress. However, there are many key conversion processes that we are not clear about. Hence, it is still necessary to further explore the genes and enzymes related to Tanshinone biosynthesis. Secondly, like most natural drugs, Tanshinone also has the characteristics of low water solubility and low bioavailability. Fortunately, the currently prepared Tanshinone injection microemulsions, microspheres, solid dispersions, liposomes, nanoparticles, and other new dosage forms can greatly improve the bioavailability of Tanshinone. However, it is still subject to the limitations of complex processes, high costs, and low tissue specificity. Therefore, further preclinical and clinical studies are needed to explore new preparations with low cost, simple processes, and high tissue specificity. It was previously reported that Tanshinone nanoparticles developed in combination with prostate-cancer-specific membrane antigen can significantly enhance the tissue specificity of Tanshinone nanoparticles for prostate cancer. Therefore, future research will continue to consider the development of drug nanoparticles in combination with tumor-specific antigens. Secondly, Tanshinone derivatives synthesized by the Tanshinone skeleton should also be a focus of future research. Although the anti-tumor effect of these derivatives is significantly lower than that of Tanshinone in most cases, some Tanshinone derivatives with increased bioavailability and anti-tumor effects have been found in the current research. Secondly, the current research on the toxic reaction of Tanshinone and Tanshinone derivatives to normal tissues is limited and cannot confirm the safety and stability of Tanshinone and its derivatives. Therefore, more attention should be paid to the study on the toxic reaction of Tanshinone and its derivatives to normal tissues in the future to further clarify the safety of Tanshinone and its derivatives. Finally, in order to better guide future research and verify the possibility of Tanshinone as an anti-prostate cancer drug, we conducted a bioinformatics analysis on the four components of Tanshinone to further determine the possible anti-prostate cancer targets of Tanshinone.

First, after determining the structural formulas of four Tanshinone components (TsI, TsIIA, DHTI, and CYT) through Pubchem (https://pubchem.ncbi.nlm.nih.gov (accessed on 10 July 2022)), we screened drug targets using the Swiss target prediction database (http://www.swisstargetprediction.ch/ (accessed on 10 July 2022)) and the traditional Chinese Medicine System Pharmacology database (https://www.tcmsp-e.com/ (accessed on 10 July 2022)), and submitted the collected targets to the UniProt database (https://www.uniprot.org/ (accessed on 10 July 2022)), limiting the species to “Homo sapiens”, converting the protein targets into official gene names, select gene targets with probability greater than 0 in the Swiss target prediction database, and obtain drug target genes: TsI (53), TsIIA (145), DHTI (80), CYT (94) after excluding duplicate genes. Secondly, we searched the Genecards (https://www.genecards.org/ (accessed on 10 July 2022)) and Disgenet databases (https://www.disgenet.org/ (accessed on 10 July 2022)) by using the keyword “prostate cancer” to obtain disease targets, and obtained 12,555 disease target genes after excluding duplicate targets in the two databases. Then, we input the drug target genes and disease target genes obtained by the above methods into the online Venny 2.1 mapping platform (https://www.bioinformatics.com.cn/ (accessed on 10 July 2022)) to obtain the cross-target genes of “prostate cancer” and “four Tanshinone components” (Figure 3), TsI (49), TsIIA (126), DHTI (68) and CYT (85) were obtained (Figure 4).

These cross genes are considered possible targets of Tanshinone against prostate cancer, and we analyzed them through a range of methods. First, we uploaded these genes to the String online database (https://string-db.org/ (accessed on 10 July 2022)) to form a protein-protein interaction map. The species is “human” and the comprehensive score > 0.4 is the critical value for inclusion in the network. We further visualized these results with the help of Cytoscape 3.9.1 (Figure 5), To find the key targets of four Tanshinone components. At the same time, we also carried out the Gene Ontology (GO) function and Kyoto Encyclopedia of Genes and Genomes (KEGG) pathway analysis. After inputting these gene data into the David data platform (https://david.ncifcrf.gov/tools.jsp (accessed on 10 July 2022)) and setting the species as “Homo species”, we further analyzed the enrichment analysis of four Tanshinone components on prostate cancer-related biological processes (BP), cellular components (CC), molecular functions (MF) and signal pathways. For the obtained information, we met the *p*-value < 0.05; sorted according to the number of genes; selected the top 10 enrichment information of BP, CC, and MF, and the top 20 enrichment information of KEGG; and used the bioinformatics online platform (https://www.bioinformatics.com.cn/ (accessed on 10 July 2022)) to visualize the analysis results (Figure 6 and Figure 7).

Finally, our results show that the target of TsIIA is significantly more than the other three components. Among them, cellular tumor antigen p53 (TP53), myc proto-oncogene protein (MYC), transcription factor AP-1 (JUN), Src, Caspase-3 (CASP3), and EGFR play a key role in the anti-prostate cancer process of TsIIA, followed by CYT, which has more targets than the other two components. EGFR, TNF, STAT3, prostaglandin G/H synthase 2 (PTGS2,), Transcription factor p65 (RELA), and other targets are the key targets of CYT, The second is DHTI, STAT3, receptor tyrosine-protein kinase erbB-2 (ERBB2), CASP3, EGFR, PTGS2, etc. are the key targets of DHTI, and finally, TsI, Vascular endothelial growth factor A(VEGFA), EGFR, MAPK14, protein tyrosine phosphatase receptor type C (PTPRC), enhancer of zeste homolog 2 (EZH2), etc. are the key targets. These key targets are essential for various biological processes of prostate cancer, and these components participate in regulating multiple signaling pathways.

Secondly, our results also show that the biological processes of the four Tanshinone components mainly occur in the cell and participate in the activation and binding of a series of cell receptors and cascade downstream signaling pathways. KEGG analysis shows that PI3K Akt signaling pathway, MAPK signaling pathway, Ras signaling pathway, and rap signaling pathway play a key role in the anti-prostate cancer process of Tanshinone, which is consistent with previous studies, These pathways have significant significance for the metastasis, progression, and angiogenesis of prostate cancer. In addition, as mentioned above, Tanshinone has a significant effect on the regulation of immune pathways. we note that the four Tanshinone components have a regulatory effect on the PD-1/PD-L1 signaling pathway, T cell-related pathways, and immune helper cells and related factors mediated signaling pathways. Although these results are not in our screening results, they are statistically different (*p*-value < 0.05). This is consistent with the protein network analysis, which may have more significant significance for immunotherapy-insensitive prostate cancer. In addition, our results also show that Tanshinone has a regulatory effect on microRNAs in prostate cancer cells, and these small molecules are also essential for tumor survival and metastasis. In conclusion, our results and the existing laboratory data show that Tanshinone can inhibit the metastasis, invasion, and progression of prostate cancer through multiple targets and pathways. However, the current research is limited. Therefore, more research should be carried out in the future to further clarify the relevant mechanisms and molecular pathways of Tanshinone against prostate cancer, and more attention should be paid to the bioavailability and toxicological experiments of Tanshinone.

## Figures and Tables

**Figure 1 molecules-27-05594-f001:**
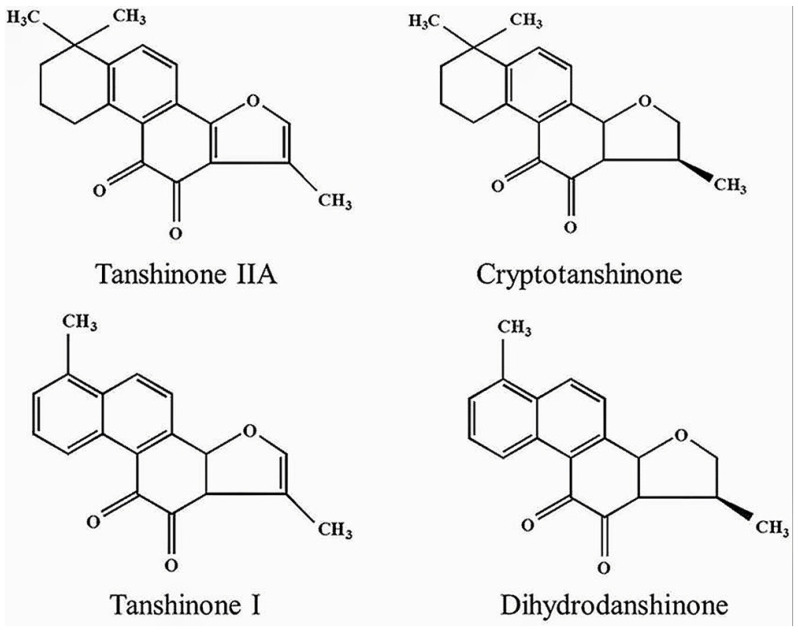
Chemical structures of four Tanshinone monomers.

**Figure 2 molecules-27-05594-f002:**
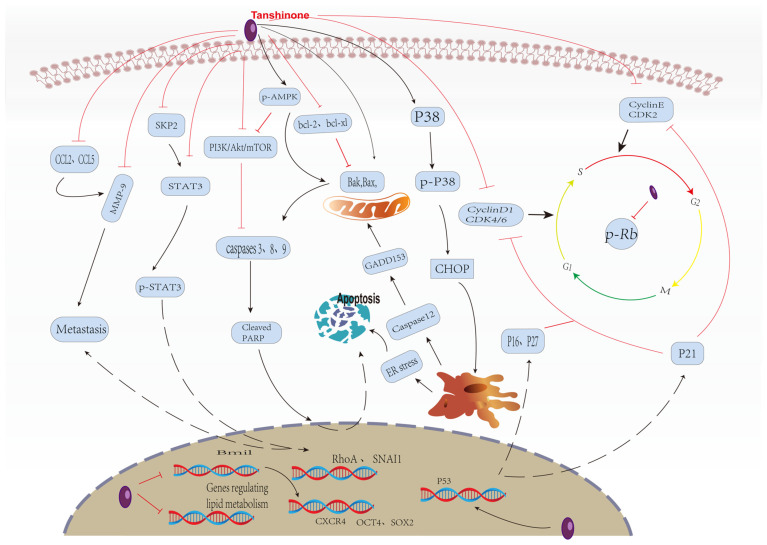
Effect of Tanshinone on prostate cancer. induced by Tanshinone are noted by using →, while the inhibition represented by ⊣ symbol.

**Figure 3 molecules-27-05594-f003:**
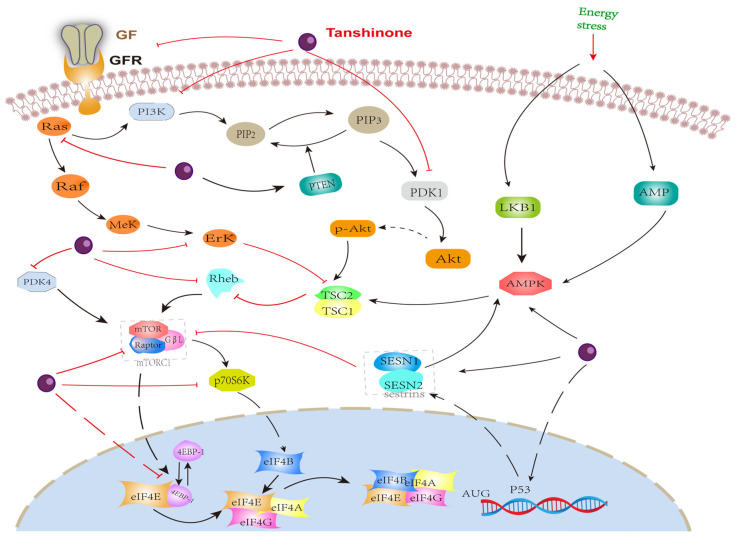
Effect of Tanshinone on mTOR. induced by Tanshinone are noted by using →, while the inhibition represented by ⊣ symbol.

**Figure 4 molecules-27-05594-f004:**
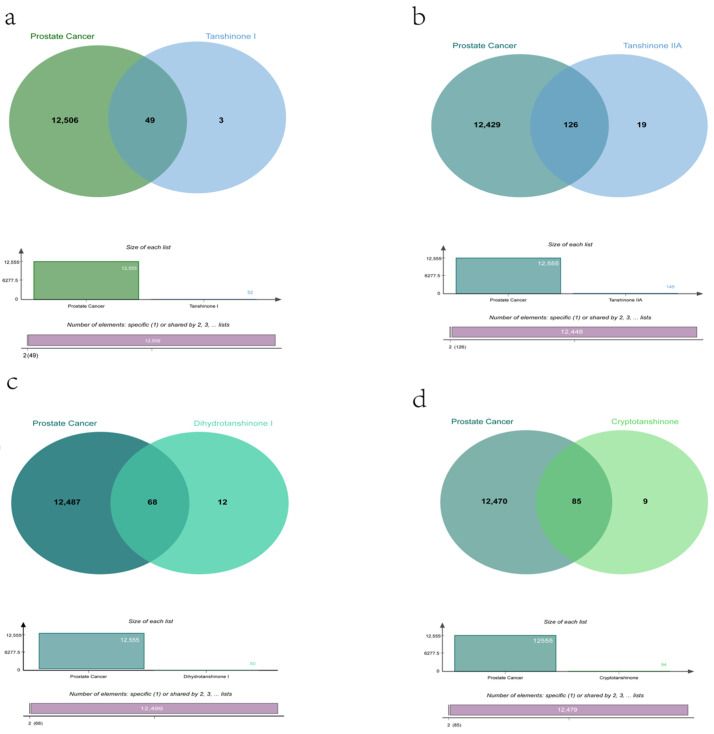
Venny of four components of Tanshinone ((**a**): Tanshinone I and Prostate Cancer; (**b**): Tanshinone IIA and Prostate Cancer; (**c**): Dihydrotanshinone I and Prostate Cancer; (**d**): Cryptotanshinone and Prostate Cancer).

**Figure 5 molecules-27-05594-f005:**
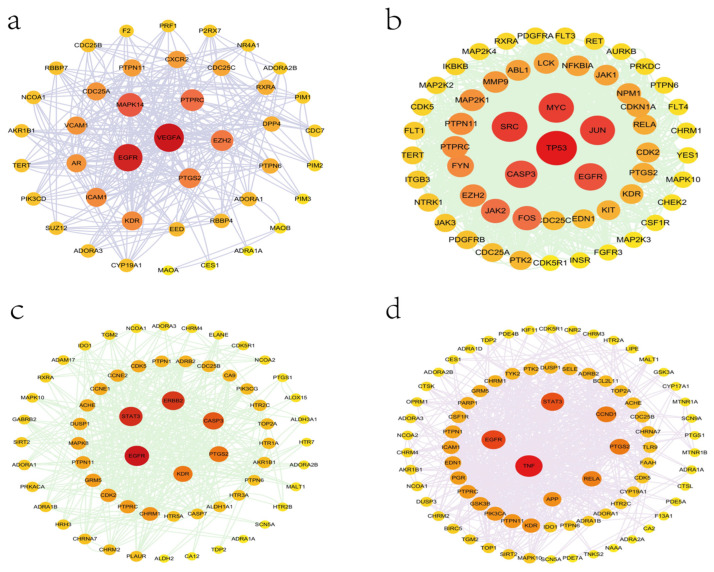
Protein network analysis of four Tanshinone components ((**a**): Tanshinone Ir; (**b**): Tanshinone IIA; (**c**): Dihydrotanshinone I; (**d**): Cryptotanshinone).

**Figure 6 molecules-27-05594-f006:**
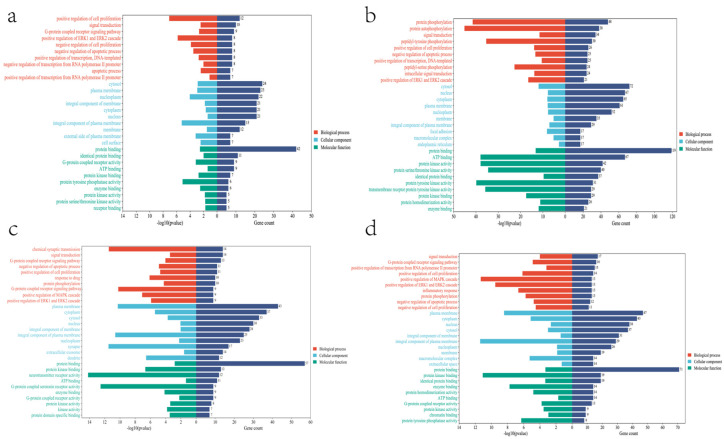
Go enrichment analysis of four components of Tanshinone((**a**): Tanshinone Ir; (**b**): Tanshinone IIA; (**c**): Dihydrotanshinone I; (**d**): Cryptotanshinone).

**Figure 7 molecules-27-05594-f007:**
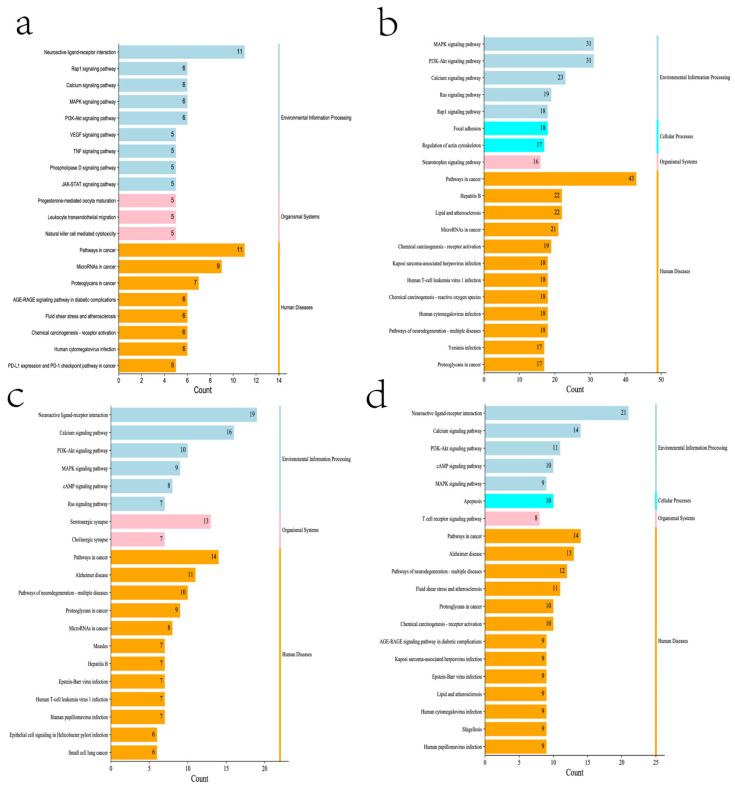
KEGG enrichment analysis of four components of Tanshinone ((**a**): Tanshinone Ir; (**b**): Tanshinone IIA; (**c**): Dihydrotanshinone I; (**d**): Cryptotanshinone).

**Table 1 molecules-27-05594-t001:** Tanshinone and PCa in vitro.

Compound	Dose	Cell	Mechanism	Reference
TsIIA	0, 1.25, 2.5, 5, 10 uM	LNCaP	Cell cycle arrest and apoptosis are induced by the activation of P53 (dose-dependent).	[34]
TsIIA	5 μM	PC-3	Induced autophagy and apoptosis	[35]
TsIIA	20 umol/L	LNCaP, PC3	enhancing the effect of the anti-tumor activity of cisplatin.	[36]
TsIIA	0, 40, 80 µM	PC-3	Inducing autophagy by up-regulated expression of microtubule-associated protein light chain 3 (LC3) II	[37]
TsIIA	10, 25, 50 uM	LNCaP, PC-3	inducing mitochondrial-dependent cell apoptosis by inhibiting PIK3/AKT	[38]
TsIIA	2.5, 5 μg/ml	LNCaP	Induced apoptosis and induced cell cycle arrest by endoplasmic reticulum stress	[39]
TsIIA	—	LNCaP	Cell proliferation was inhibited by inhibiting the AR signal.	[40]
TsIIA	—	—	Maspin expression was induced, AR expression was inhibited, and apoptosis was induced.	[41]
TsIIAD	2.5 μM	PC3	Binding NQO1 protein causes cell cycle arrest and apoptosis.	[42]
CYT	10 umol/L	DU145	Apoptosis was induced and the expression of isomucin was inhibited by inhibiting the PI3K/AKT signaling pathway.	[43]
CYT	10 μM	LNCaP, 22Rv1, and PC3	The activity and expression of AR were inhibited by inhibiting LSD1-mediated H3K9 demethylation.	[44]
CYT	1.0 ug/ml	DU145	To activate Fas-mediated apoptosis	[45]
CYT	0.5 µM	LNCaP, 22Rv1	Cell proliferation was inhibited by inhibiting AR expression and activity.	[46]
CYT	1.5 µM	LNCaP	Tumor-initiating cells are influenced by down-regulating dry gene expression.	[47]
CYT	5, 10 μM	DU145, LNCaP, and PC-3	Inhibiting HIF-1 and AEG-1 inhibits angiogenesis and induces cell cycle arrest and apoptosis.	[31]
CYT	10 μM	PC3	Cell proliferation is inhibited by decreasing the stability and expression of DNA topoisomerase 2.	[48]
CYT	5 uM	22Rv1 and PC-3	AR expression and activity were reduced, and MMP9 secretion was also reduced.	[49]
CYT	0–40 µM	DU145	Apoptosis was induced by inhibiting phosphorylation of mTOR and Rb.	[50]
CYT	7 μmol/L	DU145	Inhibition of STAT3Tyr705 and its upstream tyrosine kinase induces cell cycle arrest and apoptosis.	[51]
TsI	20, 40, 80 μM	PC-3, DU145	Apoptosis is induced by upregulation of microRNA135A-3p and death receptor 5.	[52]
TsI	3–6 μM	PC-3, LNCaP, and DU-145	inhibiting angiogenesis and inducing apoptosis by down-regulating AuroraA expression.	[30]
DHT	5–10 μM	PC-3, DU145, and 22Rv1	inhibiting EMT by inhibition of the CCL2/STAT3 axis	[34]
DHT	0.1 ug/mL and 1.5 ug/mL	DU145	Inducing cell cycle arrest by activating the ER pathway	[53]
TsD	3, 6, 12 μM	PC3, LNCAP	Inducing cell cycle arrest and apoptosis	[54]
TsD	2 µM	LNCaP, C4-2	AR expression and activity were reduced, and cell proliferation was slowed.	[55]
SME	3.125, 12.5, 25 and 50 μg/mL	DU-145	Cell cycle arrest and apoptosis are mediated by P53	[56]
SME	20 µg/ml	PC-3, LNCaP, and DU-145	Inducing cell cycle arrest and apoptosis	[57]
TsIIAN	—	PC-3 and DU145	Induction of apoptosis	[58]
SMEN	—	LNCap	Inducing apoptosis and up-regulating ROS in cells	[59]
NCDT	—	LNCaP	Enhancing toxicity of doxorubicin	[60]

**Table 2 molecules-27-05594-t002:** In animal study of Tanshinone and PCa.

Animal Models	Dose	Delivery Way	Result	Reference
22Rv1 allograft mouse model	CYT (5 mg/Kg) and CYT (25 mg/Kg)	Intraperitoneal injections were given every two days for four weeks.	Tumor growth was inhibited in both the low-dose and high-dose groups.	[46]
PC-3 allograft mouse model	TsI (150 mg/kg)	Tube feeding, once a day, for 2 weeks	Tumor weight (67%) and intratumor blood vessels (80%) were reduced.	[30]
PC-3 allograft mouse model	SME (100 mg/kg)	Oral and tube feeding, once a day, for 6 weeks	The incidence and weight of tumors were reduced.	[57]
LNCaP allograft mouse model	TsIIA (25 mg/kg)	Orally, once daily for 6 weeks	Tumor growth and the expression of AR were inhibited.	[40]
PC-3 allograft mouse model	CYT (10 mg/kg)	Intraperitoneal injection, once a day	Tumor weight (46.4%) and intratumor blood vessels were reduced.	[31]
DU-145 allograft mouse model	SME (500 mg/kg)	Orally, once daily for 2 weeks	Tumor growth was inhibited	[56]
PC-3 allograft mouse model	TsD (60 mg/kg)	Subcutaneous injections were given every two days for 18 days	Tumor growth was inhibited	[54]
LNCaP allograft mouse model	TsIIA (60 or 90 mg/kg)	Subcutaneous injections were given every two days for 13 days	Tumor weight (86.4%) was reduced.	[39]
CWR22Rv1 allograft mouse model	CYT (25 mg/kg)	Intraperitoneal injections were given 3 times per week for 4 weeks	Tumor metastasis is inhibited.	[49]
LNCaP allograft mouse model	NCDT (5 mg/Kg)	It was injected once every two days for 18 days.	To enhance the toxicity of Doxorubicin	[60]

***Note***: *Tanshinone I* (*TsI*), *Tanshinone II A* (*TsIIA*), *Tanshinone II B* (*TsIIB*), *Dihydrotanshinone I* (*DHT*), *Cryptotanshinone* (*CYT*), *Tanshinone derivatives* (*TsD*), *Tanshinone IIA derivatives* (*TIIAD*), *TsIIA nanoparticles* (*TsIIAD*), *Nanoparticles containing doxorubicin and Tanshinone* (*NCDT*), *Nanoparticles synthesized from salvia miltiorrhiza extract* (*SMED*).

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
