# Peer review of "Molecular Mechanism of Tanshinone against Prostate Cancer"

_molecules, 2022, doi:10.3390/molecules27175594_

Round 1
Reviewer 1 Report
First at all, the ms is not correctly formatted for the Journal, so it is not easy to comment it.
However, I suggest to modify the figures to better understand the role of tanshinone on prostate cancer. Moreover, I believe it will be better to focus on limitations and problems in the use of tanshinone.
Some english errors should be corrected.
Reviewer 2 Report
The review paper submitted by Li et al reviewed the available findings on the potential anticancer effect of Tanshinones against prostate cancer, the review is interesting but it requires the following modifications for improvements:
1. The manuscript requires an extensive English revision, the sentences should be properly connected, and too many typos and grammatical mistakes are there.
2. The two sections about PCa and metabolic syndrome and Tanshinone and metabolic syndrome on pages 3 and 4 should be removed as they are neither related nor justifiable.
3. A section about the differences between different Tashinones is required (the authors can discuss the differences in pharmacological characteristics, the difference in potency, doses, chemical structures, and toxicity), especially in PCa studies.
4. The authors should try to highlight and discuss the potential reasons behind the differences in doses used in both in vitro and in vivo studies for the same type of tanshinone and or cell line( for instance, some studies used 2.50 uM while other studies used 50 uM for the same tanshinone (TsIIA) on the same cell line (e.g. LNCaP))
Minor comments:
1. The authors should correct figures 2 and 3 by replacing “curcumin” with “Tanshinone”
2. NF-B should be replaced by NF-κB on the manuscript
Reviewer 3 Report
Comments:
The authors carried out this review addressing, on the one hand: 1) prostate cancer (PCa ) with some aspects of its pathophysiology, geographical distribution, signaling pathways, and incidence. The authors mention the current problems by which PCa in advanced stages can adapt and develop resistance to drugs related to androgen castration. 2) The authors focus on tanshinone and its evidence as an agent for the treatment of different types of cancer, including PCa. It is mentioned how tanshinone targets several signaling pathways involved in PCa, such as the STAT3, PI3K, AKT, mTOR, MAPK pathways, some of which are essential for the stability of this type of cancer. Therefore, in this review, the authors present the in vivo and in vitro evidence of tanshinone against prostate cancer and discuss the effect of tanshinone on nuclear factor kappa-(NF-κB), AR, and mTOR. However, some aspects must be addressed in a more synthesized and clear way within the text. The discussion of the evidence is not very clear and to a certain extent null within some paragraphs, because only a recapitulation and one real discussion is made within the mentioned data. Some points mentioned in the conclusion are not mentioned in the text, so it would be a contradiction because there is no evidence or ideas to support this section. On the other hand, there are certain problems with the format of molecules, the resolution of the images, as well as grammatical aspects. These points will be discussed extensively below.
Major observations:
Concerning the fact that the authors first mention the relationship between metabolic syndrome and PCa, and later tanshinone with PCa, this seems to be two separate ideas, no discussion relates or links these two aspects, or what would be the possible impact of tanshinone in metabolic syndrome and PCa patients, so it would be important to address this aspect.
After the section on "Tanshinone-induced apoptosis of PCa cells", it seems that there is no discussion of the evidence found, but the only mention of the processes, signaling pathways, and therapeutic targets such as NFKB, AR, mTOR, etc. In addition, the maximum number of words in the main text established by Molecules for review articles (4000 words) is exceeded, so it is important that authors carry out a more rigid synthesis of the text. A table with the most important aspects about the pathways, genes or roles involved in PCa could help to locate the reader within the review, making the text more synthesized and less tedious.
According to the molecule format, the text within the manuscript is very centered, reducing the area of the text, and further complicating its reading because it clumps together, increasing the number of pages. The format of the references in this manuscript is not MDPI format because it is displayed as a superscript. The figures 2 and 3 are not cited in the main text, and have a low resolution, and in the case of figures 2 and 3, the blue background causes some components of the track to be lost because they are also blue or have similar tones, making them difficult to read. So it is important to correct the resolution and color of the images to make them more understandable. In the captions of figures 2 and 3, the role of curcumin is not mentioned, nor a brief description about the mechanism of each pathway present in each figure. On the other hand, it would be more practical to indicate the activation and inhibition symbols within the figure and not the captions. It is important to correct these points to make the text more attractive to the reader and locate it more easily.
Within the conclusion certain points are mentioned such as the problem in the bioavailability of tanshinone and its studies related to clinical trials. However, it is not addressed within the text or discussed, so it would be important to make a section on these aspects.
On the other hand, although it is not primarily the aim of the authors, I would invite them to use network analysis in a systematic review, this idea could further enrich the work by helping to identify some targets that could share the different pathways cited in the text. , this vision would provide a better integration and a broader landscape on the mechanisms that tanshinone has on PCa.
-Sections should have the following format, for example:
1. Introduction
1.1. Current status of PCa
1.2. PCa and metabolic syndrome
.
.
.
And so on.
-No significant plagiarism was detected.
Minor observations:
The term Introduce is not correct, it should be Introduction instead. Please correct it
Salvia miltiorrhiza should be in italics (Salvia miltiorrhiza) and in vivo or in vitro.
In sections like Tanshinone AND NF-κB, AND must be lowercase.
Lane 78:Pca is spelled incorrectly, please write as PCa.
Lane 243: The idea “current vivo experiment” seems to be not suitable, please check it
Lane 442: Unknown word: tanshoneinhibition, please check it.
Lane 474-476: vcam-1, icam-1 and McP-1 must be written in uppercase, please correct it.
Lane 488: The word prostaglanins is not correct, please check it.
Lane 507: It seems that the verb rely does not agree with the subject. Consider changing the verb form.
Lane 536: But seems that conjunction use may be incorrect here.
Lane 688: This sentence ends with a double period. Consider changing the punctuation.
Reviewer 4 Report
Language verification required
Round 2
Reviewer 1 Report
I suggest to accept the ms
Author Response
Dear Reviewers:
Thank you for your letter and for the reviewers' comments concerning our manuscript entitled "Molecular Mechanism of Tanshinone against Prostate Cancer" (ID: molecules-1846760). Those comments are all valuable and very helpful for revising and improving our paper, as well as providing important guiding significance for our research.
Sincerely yours,
Kaifa Tang
Reviewer 3 Report
The authors carried out this review addressing, on the one hand: 1) prostate cancer (PCa ) with some aspects of its pathophysiology, geographical distribution, signaling pathways, and incidence. The authors mention the current problems by which PCa in advanced stages can adapt and develop resistance to drugs related to androgen castration. 2) The authors focus on tanshinone and its evidence as an agent for the treatment of different types of cancer, including PCa. It is mentioned how tanshinone targets several signaling pathways involved in PCa, such as the STAT3, PI3K, AKT, mTOR, MAPK pathways, some of which are essential for the stability of this type of cancer. Therefore, in this review, the authors present the in vivo and in vitro evidence of tanshinone against prostate cancer and discuss the effect of tanshinone on nuclear factor kappa-(NF-κB), AR, and mTOR. However, the observations given the previous time have not been fully heeded
